# The Importance of Allelopathic Picocyanobacterium *Synechococcus* sp. on the Abundance, Biomass Formation, and Structure of Phytoplankton Assemblages in Three Freshwater Lakes

**DOI:** 10.3390/toxins12040259

**Published:** 2020-04-16

**Authors:** Iwona Bubak, Sylwia Śliwińska-Wilczewska, Paulina Głowacka, Agnieszka Szczerba, Katarzyna Możdżeń

**Affiliations:** 1Division of Hydrology, Institute of Geography, University of Gdansk, Bażyńskiego 4 St, P-80-309 Gdańsk, Poland; 2Division of Marine Ecosystems Functioning, Institute of Oceanography, University of Gdansk, Avenue Piłsudskiego 46, P-81-378 Gdynia, Poland; ocessl@ug.edu.pl; 3Division of Geomorphology and Quaternary Geology, Institute of Geography, University of Gdansk, Bażyńskiego 4 St, P-80–309 Gdańsk, Poland; paulina.glowacka@phdstud.ug.edu.pl (P.G.); agnieszka.szczerba@phdstud.ug.edu.pl (A.S.); 4Institute of Biology, Pedagogical University of Krakow, Podchorążych 2 St., P-30-084 Kraków, Poland; katarzyna.mozdzen@up.krakow.pl

**Keywords:** allelopathy, inland freshwater lakes, phytoplankton assemblages, picocyanobacteria blooms, *Synechococcus* sp.

## Abstract

The contribution of picocyanobacteria to summer phytoplankton blooms, accompanied by an ecological crisis, is a new phenomenon in Europe. This issue requires careful investigation. We studied allelopathic activity of freshwater picocyanobacterium *Synechococcus* sp. on phytoplankton assemblages from three freshwater lakes. In this study, the allelopathic activity of the *Synechococcus* sp. on the total abundance, biomass, as well as structure of the phytoplankton assemblages were investigated. Our results indicated that addition of exudates obtained from *Synechococcus* sp. affected the number of cells and biomass of the phytoplankton communities; the degree of inhibition or stimulation was different for each species, causing a change in the phytoplankton abundance and dominance during the experiment. We observed that some group of organisms (especially cyanobacteria from the genus *Aphanothece*, *Limnothrix*, *Microcystis*, and *Synechococcus*) showed tolerance for allelopathic compounds produced and released by *Synechococcus* sp. It is also worth noting that in some samples, Bacillariophyceae (e.g., *Amphora pediculus*, *Navicula pygmaea*, and *Nitzschia paleacea*) were completely eliminated in the experimental treatments, while present in the controls. This work demonstrated that the allelopathic activity exhibited by the *Synechococcus* sp. is probably one of the major competitive strategies affecting some of the coexisting phytoplankton species in freshwater ecosystems. To our best knowledge this is the first report of the allelopathic activity of *Synechococcus* sp. in the freshwater reservoirs, and one of the few published works showing allelopathic properties of freshwater picocyanobacteria on coexisting phytoplankton species.

## 1. Introduction

Phytoplankton is the most numerous group of photosynthetic organisms [1], displaying a range of adaptational traits with regard to populating habitats and demonstrate a high resistance to adverse living habitats [2]. The structuring and dynamics of phytoplankton assemblages in freshwater ecosystems are driven by the relationships of phytoplankton with the chemical, physical, and biological parameters within the ecosystem [3,4,5]. Many planktonic algae, in particular blue-green algae (Cyanobacteria), are characterized by their considerable competitive potential that, in specific conditions, leads to the formation of blooms. The main reason for the forming of blooms in freshwater reservoirs is eutrophication, described as the enrichment of water bodies with nutrients [6,7,8]. Furthermore, chemically mediated interactions, so-called allelopathy, between the algal components of the aquatic ecosystem could also significantly influence phytoplankton succession [9,10].

In general, most studies of cyanobacterial and microalgal allelopathy have been performed in marine environments (e.g., [11,12,13,14,15,16]); however, little is known about this phenomenon among freshwater picocyanobacteria. In fact, only one related article on allelopathic activity in freshwater picocyanobacteria on coexisting microalgae was found. In this paper, Kovacs et al. [17] for the first time described that the freshwater picocyanobacterium *Cyanobium gracile* Rippka and Cohen-Bazire has a strong negative effect on green alga *Scenedesmus quadricauda* (Turpin) Brébisson. Most of the work on the allelopathic activity of picoplanktonic cyanobacteria of the genus *Synechococcus* has been described for the strain CCBA BA-124, originating from the Baltic Sea [18,19,20,21,22,23,24,25]. Additionally, Konarzewska et al. [26] showed allelopathic properties of Baltic *Synechococcus* sp. CCBA BA-120 and CCBA BA-132. The allelopathic activity between marine *Synechococcus* strains CC9605, CC9311, and WH8102 has also been showed by Paz-Yepes et al. [27]. There are also reports of allelopathic activity of marine picocyanobacteria of the genus *Synechocystis* CCBA MA-01 [28]. In these studies, it was shown that picocyanobacteria of the genus *Synechococcus* and *Synechocystis* can produce and release unidentified allelopathic compounds that have both a negative and positive effect on selected cyanobacteria and microalgae. The picocyanobacteria *Synechococcus* sp., *Synechocystis* sp., and *Cyanobium* sp. displayed a negative effect on the survival of selected animals [29,30,31,32]. These results obtained indicated that freshwater picocyanobacteria may serve as a potential source of interesting bioactive compounds, whose mode of action on target organisms requires detailed investigation.

Picocyanobacteria are important components and primary producers in aquatic ecosystems [33,34,35]. In the past, picocyanobacteria were described as a non-blooming group [36]; however, some studies demonstrated the potentially dangerous character of picocyanobacterial blooms [37,38,39,40]. Those bloom have caused profound transformations in the aquatic ecosystem and resulted in the loss of fish and clam resources. Furthermore, the authors examined that the picocyanobacterial bloom was accompanied by great changes in the benthic habitats. What is more, future climate change scenarios predict rising temperatures [41], which can act as a catalyst for the global expansion of harmful picocyanobacterial blooms [42,43].

As mentioned above, picoplanktonic cyanobacteria are known to produce and release a wide spectrum of biologically active compounds, whose harmfulness to other organisms has been demonstrated [44]. At the same time, the functional role of these compounds, particularly in terms of the ecology of the picocyanobacteria that produce them, remains largely unknown. Therefore, in this study, we attempted to determine the allelopathic effect of the freshwater picocyanobacterium *Synechococcus* sp. on plankton assemblages. This work emphasizes the importance of studying the allelopathic activity of picocyanobacteria in freshwater reservoirs.

## 2. Results

### 2.1. Abundance, Biomass, and Structure of the Phytoplankton Community

Altogether, 116 taxa of phytoplankton were revealed in the studied lakes: Łazduny (ŁL), Rzęśniki (RL), and Żabińskie (ŻL). A list of species detected in ŁL, RL, and ŻL in the study period is shown in Appendix A (in Appendix A). Analyzing the number of cells, it was shown that cyanobacteria dominated in all lakes throughout most of the studied depths. The highest number of cyanobacteria cells were recorded in ŻL in July (117.8∙10^6^ cell mL^–1^) and in RL in July and August (52.5∙10^6^ cell mL^–1^ and 46.4∙10^6^ cell mL^–1^, respectively).

In ŁL, Bacillariophyceae dominated in biomass in each month as well as depth, constituting about 60% of the total biomass. Biomass of the other groups was clearly lower (Figure 1A). In RL, Bacillariophyceae also dominated in biomass, constituting 42% of the total biomass of the phytoplankton assemblage and the largest share of this class was recorded from May to July at a depth of 1 m. It was also found that Chlorophyceae, Dinophyceae, and Cyanophyceae amounted to 20%, 13%, and 13% of the total biomass, respectively (Figure 1B). Bacillariophyceae also dominated in ŻL (38% of the total biomass). Furthermore, ŻL had a significant share of the Chlorophyceae (12%) and Charophyceae (21%) at a depth of 1 m (Figure 1CSurprisingly, Dinophyceae dominated in biomass in June at a depth of 1 m (56%), and of Cyanophyceae in July at a depth of 10 m (87% of biomass).

### 2.2. Effect of Picocyanobacterial Exudates on the Abundance, Biomass, and Structure of Phytoplankton Assemblages

The addition of exudates obtained from *Synechococcus* sp. affected the number of cells, biomass, as well as the structure of the phytoplankton communities (Figure 2, Figure 3 and Figure 4). It was observed that Cyanophyceae dominated in abundance in all tested samples. The only exception was the ŻL, where Chlorophyceae, Charophyceae, and Bacillariophyceae dominated in June. On the other hand, Bacillariophyceae generally dominated in biomass of the phytoplankton assemblages (Figure 2 and Figure 3).

Our study showed that the *Synechococcus* sp. exudates had a statistically significant effect on the number of Cyanophyceae cells in ŁL (Figure 2). It was shown that in May and June the Cyanophyceae cell number increased by 200% (ANOVA, *p* < 0.001) and 240% (*p* < 0.001), respectively, in relation to the control treatment. In the following months, the number of Cyanophyceae cells was significantly inhibited and amounted to 60% in July (*p* < 0.001) and 30% in August (*p* < 0.001), compared to the control samples. Similarly, the number of Bacillariophyceae cells was inhibited in June and July and stimulated in the last studied month, i.e., in August. The cell number of Bacillariophyceae in June was 55% (*p* < 0.001) relative to the control. It is worth noting that in July, Bacillariophyceae were completely eliminated in the experimental sample, while present in the control sample. In August, however, 150% (*p* < 0.001) stimulation of Bacillariophyceae growth relative to the control conditions was noted. It was shown that the exudates obtained from the *Synechococcus* sp. significantly stimulated the abundance of Euglenophyceae in July in ŁL, which was 230% (*p* < 0.001) compared to the control. In the same month, a small amount of Charophyceae was also noted in the experimental sample, which were not recorded in the control sample. In turn, the exudates from picocyanobacterium negatively affected organisms belonging to the Chrysophyceae, whose number in May was 70% (*p* < 0.05) compared to the control, and the number of Chlorophyceae, whose number was 75% in June (*p* < 0.001), 60% (*p* < 0.001) in July, and only 5% (*p* < 0.001) in August.

It was noted that *Synechococcus* sp. exudates significantly inhibited the cell numbers of Cyanophyceae in RL in all the studied months (Figure 2). The numbers of Cyanophyceae cells in May, June, July, and August were 70% (*p* < 0.001), 70% (*p* < 0.001), 90% (*p* < 0.001), and 70% (*p* < 0.001), respectively, compared to the control samples. It was also shown that the exudates obtained from *Synechococcus* sp. had a negative effect on the number of cells of Bacillariophyceae and Charophyceae in June. Their overall number of cells in the experimental sample was 80% (*p* < 0.001) and only 5% (*p* < 0.001), respectively, compared to the control treatment. It is worth noting that in May and July, Bacillariophyceae were completely eliminated in the experimental sample, while present in the controls. In addition, it was found that the population of Bacillariophyceae was, similarly to that recorded in ŁL, stimulated by 270% in August (*p* < 0.001). It was also shown that the exudates from picocyanobacterium stimulated the abundance of Chlorophyceae in RL. In June, July, and August, the number of Chlorophyceae cells increased by 130% (*p* < 0.001), 150% (*p* < 0.001), and about 480% (*p* < 0.001), respectively, compared to the control treatment.

The picocyanobacterium *Synechococcus* sp. exudates had a statistically significant effect on the cell number of Cyanophyceae in ŻL (Figure 2). In May and June, as in the case of ŁL, the number of Cyanophyceae increased statistically and amounted to 115% (*p* < 0.01) and 200% (*p* < 0.001), respectively, in relation to the control. In turn, in the following months, the number of Cyanophyceae cells was significantly inhibited and amounted to only 10% (*p* < 0.001) in July and 60% (*p* < 0.001) in August, compared to the control samples. *Synechococcus* sp. exudates significantly inhibited the number of Bacillariophyceae cells in ŻL in all the studied months. The cells numbers of Bacillariophyceae in May, June, July, and August were as follow: 80% (*p* < 0.001), 10% (*p* < 0.001), 85% (*p* < 0.01), and 50% (*p* < 0.001), respectively, compared to the control sample. It was also shown that the exudates from the picocyanobacterium generally stimulated the number of Chlorophyceae and Charophyceae cells in ŻL. In June, July, and August, the number of Charophyceae cells increased by 160% (*p* < 0.001), 210% (*p* < 0.001), and about 600% (*p* < 0.001), respectively, compared to the control. In contrast, stimulation of Chlorophyceae was 210% in May (*p* < 0.001), 140% in June (*p* < 0.001), and 200% in August (*p* < 0.001). Furthermore, in May, a small amount of Cryptophyceae was noted in the experimental sample while in the control sample they were not recorded.

It was shown that the exudates from *Synechococcus* sp. significantly reduced the biomass of Bacillariophyceae in May, June, and July in ŁL (Figure 3A). Moreover, it is worth noting that in May and July, Bacillariophyceae were completely eliminated in the experimental samples, while present in the controls. In May, it was found that the *Synechococcus* sp. exudates caused stimulation of Cyanophyceae biomass. Their biomass was about 3550% compared to the control sample. Furthermore, it was found that Cyanophyceae: *Aphanocapsa holsatica* (Lemmermann) G.Cronberg and Komárek, *Aphanocapsa incerta* (Lemmermann) G.Cronberg and Komárek, and *Snowella atomus* Komárek and Hindák were recorded in both the control and experimental samples. However, the species that were present only in the experimental sample were *Aphanothece* sp. and *Woronichinia naegeliana* (Unger) Elenkin (Appendix A in Appendix A). In August, in turn, it was shown that the picocyanobacterial exudates caused stimulations of Bacillariophyceae, whose biomass was 210% compared to the control. In the same month, there was a decrease in biomass of Chlorophyceae in the experimental sample. Species that dominated in the control were *Coelastrum microporum* Nägeli, *Monoraphidium contortum* (Thuret) Komárková-Legnerová, *Monoraphidium minimum* (Nägeli) Komárková-Legnerová, and *Tetrastrum staurogeniaeforme* (Schröder) Lemmermann, while only *M. minutum* was recorded in the experimental sample.

It was found that higher overall phytoplankton biomass was observed in the RL in the control in May and July. In these months, as was the case in ŁL, it was noted that Bacillariophyceae were completely eliminated in the experimental samples, compare to the control samples. Surprisingly, in June an increase in the overall biomass of Bacillariophyceae was noted in the experimental treatment, namely *Amphora pediculus* (Kützing) Grunow, *Navicula* sp., and *Pinnularia* sp. (Appendix A). Their biomass was almost 200% compared to the control sample. In August, it was shown that the picocyanobacterial exudates caused stimulations of Bacillariophyceae and Chlorophyceae, whose biomass was 170% and 520%, respectively, compared to the control treatment (Figure 3B).

Generally, in ŻL, the exudates from *Synechococcus* sp. had a negative effect on the biomass of phytoplankton assemblages throughout the entire studied period (Figure 3C80% (*p* < 0.01), 10% (*p* < 0.001), 80% (*p* < 0.01), and 20% (*p* < 0.001), respectively, relative to the control treatments. Species that were present in the control but were eliminated in the experimental sample were *Achnanthes* sp., *A. pediculus*, *Navicula pygmaea* Kützing, and *Nitzschia paleacea* (Grunow) Grunow (Appendix A). In May, June, and August it was noted that the picocyanobacterial exudates caused stimulations of Chlorophyceae, whose biomass was 180%, 160%, and 160%, respectively, compared to the controls. The species that dominated in both the control and experimental samples were *Desmodesmus communis* (E.Hegewald) E.Hegewald, *M. contortum*, *M. minutum*, and *Tetraëdron minimum* (A.Braun) Hansgirg (Appendix A). It was also shown that the exudates from *Synechococcus* sp. in June, July, and August caused stimulations of Charophyceae, whose biomass was 160%, 200%, and 610%, respectively, compared to the control treatments. It is worth mentioning that the only species recorded in the samples was *Koliella longiseta* (Vischer) Hindák (Appendix A).

We also showed that different phytoplankton species responded differently to *Synechococcus* sp. exudates (Figure 4). It was found that Cyanophyceae from the genus *Aphanothece*, *Limnothrix*, *Microcystis*, *Planktolyngbya*, *Pseudanabaena*, *Synechococcus*, and *Woronchininia*, as well as Chlorophyceae and Charophyceae (*Ankistrodesmus*, *Cosmarium*, *Dictyosphaerium*, *Pediastrum*, *Planktonema*, and *Scenedesmus*) showed tolerance for allelopathic compounds produced and released by freshwater *Synechococcus* sp. in each lake. In our work we have shown that *Synechococcus* sp. also stimulated the growth of some species of Bacillariophyceae, especially from the genus *Odontella* and *Stauroneis*. On the other hand, *Chroococcus* and *Lemmermaniella* (Cyanophyceae), *Sphaerocystis* and *Koliella* (Chlorophyceae), as well as *Achnanthes*, *Amphora*, *Gomphonema*, and *Nitzschia* (Bacillariophyceae) were strongly inhibited by this picocyanobacterium.

## 3. Discussion

We showed that phytoplankton assemblages responded differently to *Synechococcus* sp. allelopathy. In this work we demonstrated that these differences in susceptibility are found between major taxonomic groups and even between individual species. This differential effect on microalgae species suggests that picocyanobacterial exudates have an important role in structuring phytoplankton assemblages.

The allelopathic activity between cyanobacteria that occur in the same ecosystem is an interesting concept in terms of evolution. The literature data indicated that some Cyanophyceae could produce the allelopathic compounds that affect the growth of other cyanobacterial species (e.g., [45,46,47,48,49,50,51,52,53]). What is more, recent research showed that picoplanktonic cyanobacteria are also capable of allelopathic effects on other cyanobacteria. Paz-Yepes et al. [27] used liquid and plate assays to demonstrate *Synechococcus* sp. inhibited growth of other *Synechococcus* sp. strains. Barreiro Felpeto et al. [25] demonstrated that *Synechococcus* sp. also had a strong inhibitory effect on *Nodularia spumigena* Mertens ex Bornet and Flahault and, surprisingly, there was no target organism reciprocal effect. Śliwińska-Wilczewska et al. [20] described the adverse impact of *Synechococcus* sp. filtrate against *Nostoc* sp. and *Phormidium* sp. Moreover, the authors showed that the addition of picocyanobacterial filtrate stimulated the growth of *Aphanizomenon flos-aquae* Ralfs ex Bornet and Flahault and had no allelopathic effects on *Rivularia* sp. Śliwińska-Wilczewska et al. [21] also indicated that the degree of inhibition was different for each species, causing a change in the phytoplankton abundance and dominance during the experiment. The authors demonstrated that the picocyanobacterium *Synechococcus* sp. filtrate generally had an inhibitory effect on the phytoplankton community, except for the cyanobacteria *N. spumigena* and *Gloeocapsa* sp., which increased in the filtrate treatment. Recent studies have also shown that *Synechococcus* sp. had a strong inhibitory effect on other cyanobacteria from the genus *Phormidium*, *Planktolyngbya*, *Pseudanabaena*, *Nostoc*, and *Synechocystis* sp. while stimulating *Aphanizomenon* sp. [26]. It is still not understood precisely why cyanobacteria produce compounds that perform stimulatory activity. Some researchers believe that cyanobacteria are capable of secreting some autostimulators that accelerate the development of the same species in the environment [54]. Moreover, it is commonly known that in laboratory experiments using monocultures, generally, cyanobacteria inhibit the growth of other cyanobacteria [55,56]; however, in natural assemblies, many co-occurring species could have developed some protective mechanisms against cyanobacterial metabolites and even benefit from them [57]. Our observations indicated that some Cyanophyceae (especially cyanobacteria from the genus *Aphanothece*, *Limnothrix*, *Microcystis*, and *Synechococcus*) may show tolerance for allelopathic compounds produced and released by freshwater *Synechococcus* sp., which may be the result of coevolution during their coexistence in some freshwater ecosystem. It is worth noting here that mutual stimulation of picoplanktonic cyanobacteria may indicate their competitive advantage and explain their high abundances in the summer season in some freshwater reservoirs.

Studies have shown that cyanobacteria can also affect the growth of some Chlorophyceae and Charophyceae species [13,50,51,52,58,59,60,61,62,63] and picoplanktonic cyanobacteria deserved special attention here. Śliwińska-Wilczewska et al. [22] demonstrated that both the addition of *Synechococcus* sp. cell-free filtrate and co-culture inhibited the growth of *Stichococcus bacillaris* Nägeli. Moreover, Śliwińska-Wilczewska and Latała [23] and Śliwińska-Wilczewska et al. [24] noted that *Synechococcus* sp. also inhibited the growth of *Chlorella vulgaris* Beyerinck [Beijerinck]. Recently, Kovács et al. [17] demonstrated that the freshwater picocyanobacterium *C. gracile* had a substantial negative impact on the coexisting *S. quadricauda*. Konarzewka et al. [26] also showed strong inhibition of Chlorophyceae growth due to exudates from three different *Synechococcus* phenotypes. Similar observations have been also made for other picocyanobacterium *Synechocystis* sp. [28]. Contrary to that, *Synechococcus* sp. filtrate had no allelopathic effects on *Oocystis submarina* Lagerheim [23,24]. In our work we have shown that *Synechococcus* sp. also stimulated the growth of some Chlorophyceae and Charophyceae (e.g., *Ankistrodesmus*, *Cosmarium*, *Dictyosphaerium*, and *Pediastrum*). Those results may indicate that cyanobacteria are capable of producing more than one bioactive compound that affect different target organisms. Allelopathic effects recognized in cyanobacteria may play an important role in the deterrence of target organisms from colonization of cyanobacteria cells [64]. Our findings suggest that allelopathic compounds secreted by the picocyanobacterium *Synechococcus* sp. may be responsible for their natural selection and ecological succession by inhibiting co-occurring competitive Chlorophyceae and Charophyceae species, especially from the genus *Sphaerocystis* and *Koliella*. This work also demonstrated that freshwater picocyanobacterium can affect the phytoplankton community differently, depending on the coexisting species.

Bacillariophyceae seem to be very sensitive to allelopathic compounds; some studies documented the allelopathic effect of cyanobacteria on selected diatoms species [45,59,65,66,67]. Picoplanktonic cyanobacteria may also affect the occurrence of Bacillariophyceae (Figure 4). Śliwińska-Wilczewska et al. [18] described that the picocyanobacterium *Synechococcus* sp. affected coexisting diatom *Navicula perminuta* Grunow negatively; it was the first of such a report in the literature. One year later, Śliwińska-Wilczewska et al. [21] examined the influence of allelopathic compounds on the growth, total abundance, and composition of a phytoplankton community by adding the cell-free filtrate of *Synechococcus* sp. into the medium. That study pointed to the diatoms of the genera *Navicula*, *Chaetoceros*, *Amphora*, *Coscinodiscus*, *Grammatophora*, and *Nitzschia* as the most allelochemical-sensitive organisms. Moreover, Śliwińska-Wilczewska and Latała [23], Śliwińska-Wilczewska et al. [24], and Konarzewska et al. [26] demonstrated that the addition of *Synechococcus* sp. filtrate strongly inhibited the growth of *Skeletonema marinoi* Sarno and Zingone. In contrast, Śliwińska-Wilczewska et al. [22] showed that *Nitzschia dissipata* (Kützing) Rabenhorst was not affected by the picocyanobacterial filtrate or co-culture. It was also found that marine *Synechocystis* sp. was able to inhibit *Fistulifera* sp. growth [28]. The susceptibility of target Bacillariophyceae to allelochemicals may depend on the nature of allelopathic compounds to which they are exposed, because the same target organisms may responsd differently to the exudates obtained from different donor organisms. Additionally, some co-evolutionary aspects may contribute to the observed results [45]. Diatom blooms usually do not co-exist with the massive cyanobacterial blooms. Therefore, in a natural environment, Bacillariophyceae generally do not have the opportunity to develop any defense mechanism for the allelopathic compounds secreted by cyanobacteria; it is likely a reason of the picocyanobacterial, allelopathic, effect-driven significant inhibition of diatom growth (especially from the genus *Achnanthes*, *Gomphonema*, and *Nitzschia*) in the studied freshwater reservoirs. It is worth mentioning here that most of the cyanobacterial allelochemicals are still unknown. Therefore, demonstrating which of the allelopathic compounds of *Synechococcus* sp. are responsible for the observed effects requires further, detailed research.

This work clearly demonstrates that the allelopathic activity exhibited by the *Synechococcus* sp. is probably one of the major competitive strategies affecting some of the coexisting phytoplankton species in freshwater ecosystems. Coevolution is hypothesized to be the main reason explaining the differences in phytoplankton susceptibility to picocyanobacterial exudates. Our observations indicated that (i) some phytoplankton species may show tolerance for allelopathic compounds produced and released by freshwater *Synechococcus* sp., which may be the result of coevolution during their coexistence in some freshwater ecosystem; (ii) mutual stimulation of picocyanobacteria may explain their high abundances in the summer season in some freshwater reservoirs; and (iii) the allelopathic effect may be dependent on the specificity of the target group and season. To our best knowledge this is the first report of the allelopathic activity of *Synechococcus* sp. in freshwater reservoirs, and one of the few published works showing allelopathic properties of freshwater picocyanobacteria on coexisting phytoplankton species. Therefore, to fully understand the allelopathic effects in aquatic environments, studies on different phytoplankton assemblages in many freshwater ecosystems are still needed to be performed.

## 4. Materials and Methods

### 4.1. Study Sites

The studied lakes are located in northeastern Poland in the Masurian Lakeland (Figure 5). This region is characterized by well-preserved postglacial landforms with the highest areal density of lakes in Poland [68]. The investigated lakes are small and relatively deep, but they are different in terms of morphometry, trophic status, hydrological regime, and catchment size (Table 1).

Lake Żabińskie (ŻL) (54°07′54.2″ N, 21°58′56.5″ E, 117 m asl) is the largest (Table 1) and occupies a glacially eroded depression formed during Vistulian glaciation (ca. 15.2 kaBP) [69]. The total catchment has a surface of 24.6 km^2^ and it is mostly covered by forests (65%). ŻL has three inflows, a major one from the northeastern side (from Lake Purwin) and two smaller creeks flowing from the south and southeastern side. The outflow drains water to the larger lake Gołdopiwo located in the west (Figure 5). Carlson’s [70] Trophic State Index (TSI) (determined on the base of chlorophyll-*a* concentration, total phosphorus concentration and Secchi disc transparency) indicate the eutrophic status of ŻL.

Lakes Łazduny (ŁL) (53°51′18.3″ N, 21°57′07.1″ E, 128.8 m a.s.l.) and Lake Rzęśniki (RL) (53°50′30.0″ N, 21°58′35.9″ E, 125.3 m a.s.l.) were formed probably by the melting of dead ice in a deep channel of a glacial outwash plain. The lakes have a common catchment (total surface area of 1.94 km^2^) covered by coniferous forests (85%) and also have similar basic morphometric parameters (Table 1). ŁL is an outflow lake, from which waters flows through two small and shallow water bodies and enters RL in its northwestern part. Outflow from RL supplies water to the Lake Orzysz located in the south-east (Figure 5). The TSI of ŁL and RL indicate the mezotrophic lake state of both.

### 4.2. Determination of Abundance, Biomass, and Structure of the Phytoplankton Community

Samples of phytoplankton were collected from ŁL, RL, and ŻL with 1-month intervals from May to August 2019 (Table 2). Samples were taken from the 1 m and 10 m depth. During field tests, the physical and chemical parameters (temperature, conductivity, pH, oxygen saturation and concentration, turbidity) of the water column were measured using a multiparameter sonde YSI 6820 meter (YSI, Yellow Spring, USA). The concentrations of nutrients (TN and TP) were measured using colorimetric methods and a Spectroquant NOVA 400 spectrophotometer (Merck, Darmstadt, Germany).

Plastic bottles were used to collect phytoplankton. The collection and processing of materials was carried out according to generally accepted methods in algology—water in each carboy was mixed, then a 500 mL sample was taken from each carboy and fixed with 5 mL of Lugol’s solution. The pre-labeled sample bottles were transported to the laboratory for analysis [72]. International handbooks were used to identify algae [2,73], and the taxa names were adopted to the international system in Algaebase [74].

For quantitative analysis, including the identification, enumeration, and calculation of biovolumes of Lugol’s iodine preserved water samples, Utermöhl’s inverted-microscope method was applied. The preserved samples were thoroughly mixed, and a sub-sample of known volume was placed in a sedimentation chamber of 5 to 100 mL capacity (Utermöhl’s plankton chambers or similar are recommended). Cyanobacteria and algae belonging to the phytoplankton community were counted and identified using an inverted microscope (Nikon, Japan). The average number of individuals/transect was converted into a population density unit (ind. mL^–1^). Population density data were translated into phytoplankton biomass by taking into account differences in the cell size of a particular taxa. For the calculation of the specific cell volumes, simple geometric models (e.g., sphere, ellipsoid, cylinder) were used [75].

### 4.3. Determination of the Allelopathic Activity of Synechococcus Exudates

Determination of the allelopathic activity of *Synechococcus* sp. exudates on phytoplankton assemblages was tested according to the methods described by Śliwińska-Wilczewska et al. [21]. The experiments were conducted on the freshwater picocyanobacterium *Synechococcus* sp. (CCBA AR-258, Figure 6). This specific strain of picocyanobacteria was used in this study due to its previously detected allelopathic activity (data not shown). This strain was maintained as unispecies cultures in the Culture Collection of Baltic Algae (University of Gdańsk, Poland). *Synechococcus* sp. was grown in f/2 culture medium [76] in 100 mL glass flasks. Culture media was prepared with distilled water and autoclaved (15 min, 121 °C). The picocyanobacterial culture was kept in a culture room at 20 °C with a 16:8 h light:dark cycle at 50 μmol photons m^–2^s^–1^ of photosynthetically active radiation (PAR). The phytoplankton communities used for the experiment was taken from the same lakes and at the same time as the phytoplankton used for the biological analysis: for species composition and biomass calculation, see Subsection 4.2. The samples were filtered through a 150 μm mesh-size nylon net to remove the effect of grazing by mesozooplankton. The phytoplankton assemblages used in the experiments were kept under a temperature of 20 °C and light intensity of 50 μmol photons m^–2^s^–1^ (PAR) for 21 days before the experiments to acclimatize the collected material to these conditions.

The allelopathic activity was estimated by adding a specific volume (10 mL) of the exudates obtained from donor picocyanobacterial culture to the phytoplankton assemblages (10 mL) kept in 25 mL glass flasks. Controls consisted of the addition of 10 mL of f/2 medium to the 25 mL flasks containing 10 mL of the phytoplankton assemblages. The nutrient level was tested to make sure that it was the same as in the portions of fresh f/2 medium added in the control samples according to the methodology proposed by Śliwińska-Wilczewska and Latała [23]. The *Synechococcus* sp. culture was gently filtered through a 0.45-µm membrane filter (Macherey-Nagel, Germany) using a vacuum pump. The cell abundance in the donor *Synechococcus* sp. cultures was 10^6^ mL^–1^. This concentration was selected to represent the appropriate environmental conditions, and high enough to be measured properly. The exudates were analyzed on an epifluorescence microscope (Nikon Eclipse 80i, Japan) to confirm the absence of the picocyanobacteria cells. The time of the experimental phase was 1 week, and all treatments were analyzed in independent triplicates.

### 4.4. Statistical Analyses

One-way ANOVA was used to test the effect of picocyanobacterial exudates on the number of cells and biomass of the targeted cyanobacteria and microalgae on the last day of the experiment. Data are reported as means ± standard deviations (SD). Levels of significance were * *p* < 0.05; ** *p* < 0.01; *** *p* < 0.001. The statistical analyses were performed using Statistica^®^ 13.1 software.

## Figures and Tables

**Figure 1 toxins-12-00259-f001:**
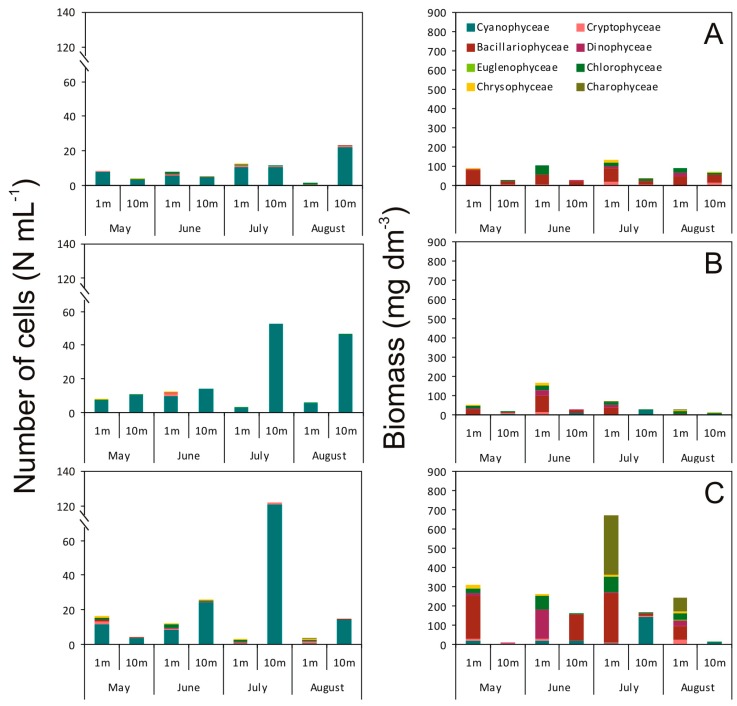
The number of cells (10^6^ cell mL^–1^) and biomass of each phytoplankton class in the Łazduny (ŁL) (**A**), Rzęśniki (RL) (**B**), and Żabińskie (ŻL) (**C**) lakes at a depth of 1 m and 10 m during the study period.

**Figure 2 toxins-12-00259-f002:**
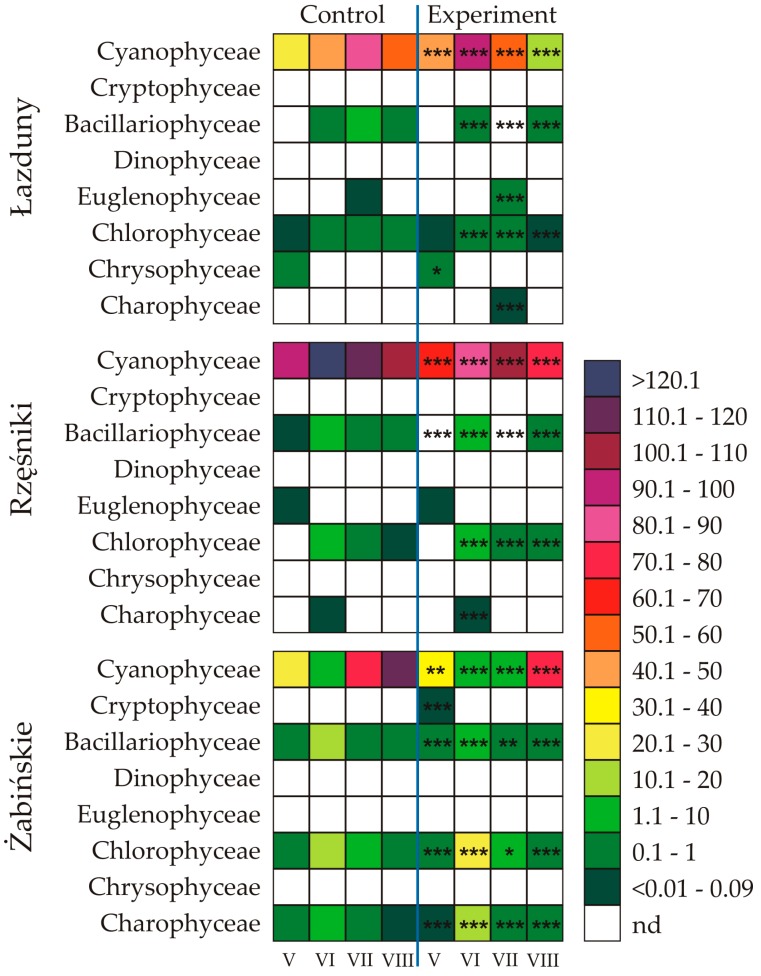
List of class of phytoplankton in the studied lakes, the number of cells (10^6^ cell mL^-1^) of species in the controls, and the experiments after 7 days of exposition to the exudates from *Synechococcus* sp. measured for each month (*n* = 3, mean ± SD). Asterisks indicate statistically significant difference compared with the control (ANOVA test, * *p* < 0.05; ** *p* < 0.01; *** *p* < 0.001; nd—not detected).

**Figure 3 toxins-12-00259-f003:**
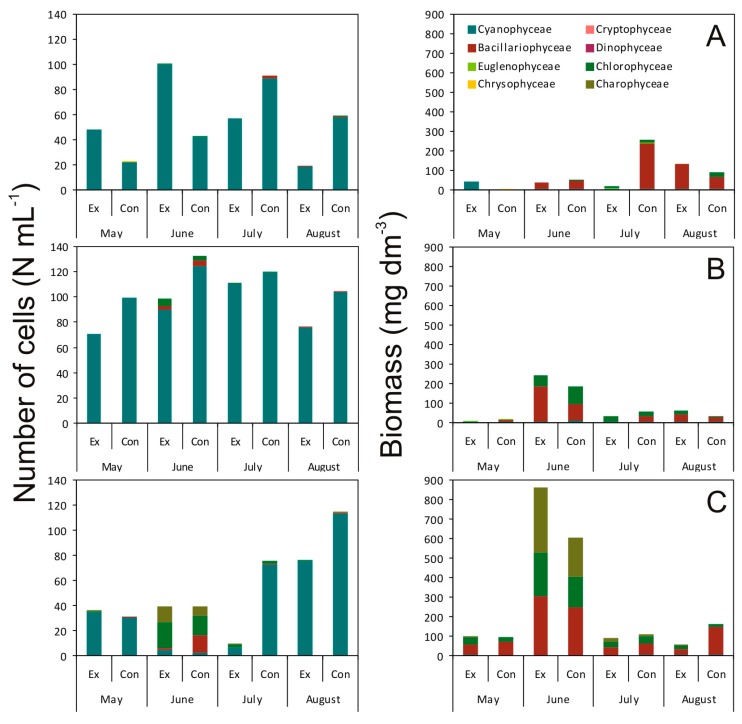
The number of cells (10^6^ cell mL^–1^) and biomass of taxa present in the ŁL (**A**), RL (**B**), and ŻL (**C**) lakes in the control (Con) and the experimental samples (Ex) measured during the study period (*n* = 3).

**Figure 4 toxins-12-00259-f004:**
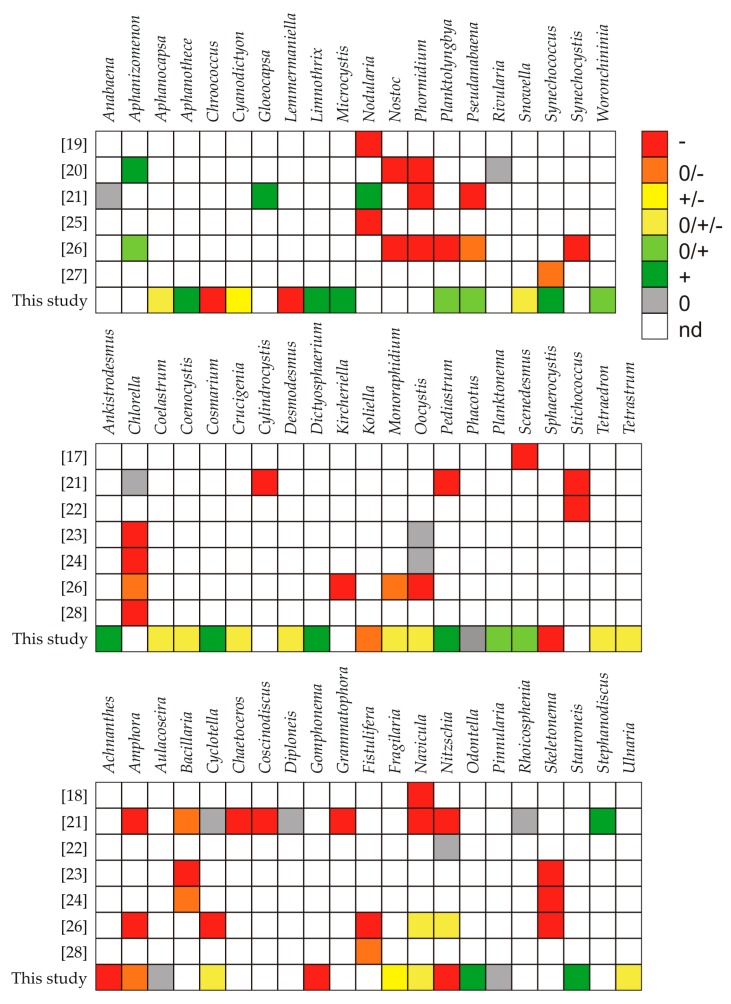
List indicating the allelopathic activity of picocyanobacteria against other genera of cyanobacteria and microalgae with the references (− means inhibiting effects, + means stimulating effects, and 0 means a lack of an effect; nd—not detected).

**Figure 5 toxins-12-00259-f005:**
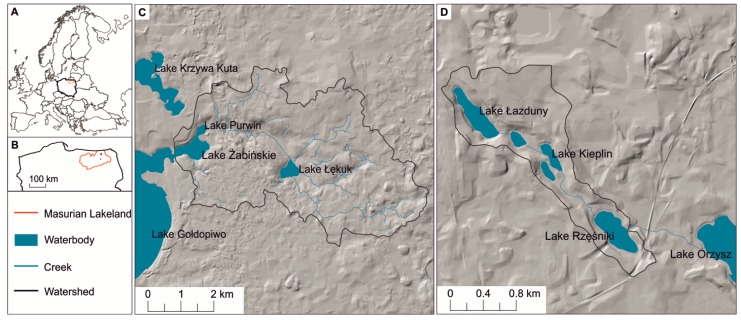
Localization of the studied lakes in Masurian Lakeland, north-eastern Poland. (**A**) Poland in Europe; (**B**) the Masurian Lakeland in Poland; (**C**) the watershed of LŻ; and (**D**) the watersheds of LŁ and LR.

**Figure 6 toxins-12-00259-f006:**
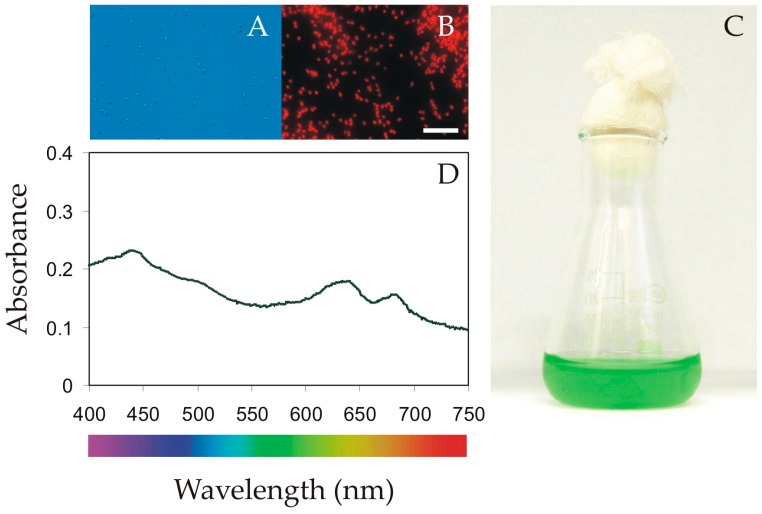
Light (**A**) and epifluorescence (**B**) microscope photographs of picocyanobacteria strain CCBA AR-258 (scale = 10 µm); photographs of the picocyanobacterial culture in 100 mL glass flasks from the experimental phase (**C**); and PAR absorption spectra determined for this strain at an optical density (OD_750_) = 0.1 (**D**).

**Table 1 toxins-12-00259-t001:** Characteristic features of the studied lakes and their catchments [71].

Parameter	Lake		
	Łazduny	Rzęśniki	Żabińskie
Surface (ha)	10.6	12.0	41.6
Volume (tys m^2^)	964.6	1111.8	5072.8
Maximum depth (m)	22.4	26.0	44.4
Average depth (m)	9.1	7.8	12.2
Maximum length (m)	790	700	1073
Maximum width (m)	210	280	635
Length of shoreline (m)	1880	1700	2846
Shoreline development index	1.6	1.3	1.2
Exposure index	3.4	1.8	3.4
Hydrological type	outflow lake	flow lake	flow lake
Total surface of catchment (km^2^)	1.94	1.94	24.6

**Table 2 toxins-12-00259-t002:** Physicochemical parameters of the water of the studies lakes during the study period.

Lake	Month	Depth [m]	T (°C)	pH	EC (µS·cm^-1^)	TP (mg P·dm^-3^)	TN (mg N·dm^-3^)
Łazduny	May	1	12.9	8.5	383	0.08	0.04
		10	5.4	7.5	394	0.07	0.38
	June	1	23.3	8.5	378	0.03	0.64
		10	5.3	7.4	397	0.06	1.41
	July	1	23.4	8.6	375	0.17	0.79
		10	6.4	7.6	396	0.05	1.08
	August	1	22.5	8.5	369	0.10	0.62
		10	6.4	7.7	398	0.26	0.67
Rzęśniki	May	1	13.2	8.3	392	0.03	0.41
		10	4.7	7.5	406	0.09	1.05
	June	1	23.2	8.3	384	0.01	0.78
		10	4.8	7.4	406	0.05	0.82
	July	1	23.8	8.4	383	0.07	0.80
		10	5.1	7.5	406	0.29	1.21
	August	1	22.8	8.4	372	0.08	0.58
		10	5.3	7.6	407	0.12	0.90
Żabińskie	May	1	13.0	8.8	370	0.06	2.07
		10	6.0	7.6	432	0.07	1.60
	June	1	-	-	-	0.09	1.24
		10	-	-	-	0.22	2.30
	July	1	23.7	8.8	325	0.06	1.14
		10	6.1	7.6	436	0.18	2.39
	August	1	22.7	8.7	340	0.16	0.59
		10	6.1	7.6	438	0.23	2.04

Note: T—water temperature; EC—electric conductivity; TP—total phosphorus; TN—total nitrogen.

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
