# Peer review of "The Importance of Allelopathic Picocyanobacterium Synechococcus sp. on the Abundance, Biomass Formation, and Structure of Phytoplankton Assemblages in Three Freshwater Lakes"

_toxins, 2020, doi:10.3390/toxins12040259_

Round 1

Reviewer 1 Report

The paper has a certain value for the reader but it has some deficiencies that shuld be dealt with befor its publishing.

What was the rationale for using this specific cyanobacterial strain in this study?

Did the authors check the remaining nutrient content in the "exudates" (which was actually the culture broth, not really exudate) to make sure that it was the same as in the portions of fresh f/2 medium added in control?

Fig. 1: left column - duplicating letters in the panel annotations are confusing; right column - I did not encounter these units as units of (bio)mass.

Table 1: it is very long and difficult to read. Can it be represented e.g. as a heatmap?

Discussion is very long record of ample facts from interactions of cyanobacteria with each other and microalgae. it should be shortened e.g. by converting some fact into a figure e.g. with a matrix. The species (maybe with the corresponding references) can be listed on X and Y dimensions and the interaction type (++, +- or --) can be indicated on the intersections.

LL438-439: incomplete sentence?

Author Response

Response to Reviewer 1 Comments

The paper has a certain value for the reader but it has some deficiencies that should be dealt with before its publishing.

Response: The authors would like to thank Reviewer 1 for her/his helpful and constructive comments that will greatly contribute to the improvement of our paper. We have studied your comments carefully and have made a correction. We hope the revised version will be satisfactory. All the modifications in the manuscript are marked in green color.

General comments:

Point 1: What was the rationale for using this specific cyanobacterial strain in this study?

Response 1:

According to reviewer suggestion, this issue was added.

Revised paragraph (Lines 448-449):

This specific strain of picocyanobacteria was used in this study due to its previously detected allelopathic activity (data not shown).

Point 2: Did the authors check the remaining nutrient content in the "exudates" (which was actually the culture broth, not really exudate) to make sure that it was the same as in the portions of fresh f/2 medium added in control?

Response 2:

Thank you for that comment. We want to assure you that the level of nutrients is always examined in our works [e.g., Śliwińska-Wilczewska et al., 2017, 2018, 2019; Śliwińska-Wilczewska and Latała, 2018]. In this work, the nutrient level was also tested to make sure that it was the same as in the portions of fresh f/2 medium added in control. We added this information to the text.

Revised paragraph (Lines 463-465):

The nutrient level was tested to make sure that it was the same as in the portions of fresh f/2 medium added in control samples according to the methodology proposed by Śliwińska-Wilczewska and Latała [25].

Point 3: Fig. 1: left column - duplicating letters in the panel annotations are confusing; right column - I did not encounter these units as units of (bio)mass.

Response 3:

We corrected Fig. 1. We hope the revised version of Fig. 1 will be satisfactory. The biomass unit used in the manuscript refers to the biovolume of phytoplankton (it is the one of way to represent biomass). This is the method recommended by institutions dealing with environmental monitoring, e.g. HELCOM or Inspectorate for Environmental Protection in Poland (for example Guidelines-for-monitoring-phytoplankton-species-composition-abundance-and-biomass.pdf). However, in typical scientific publications is used the different unit: mg·dm-3. Therefore, the authors of this publication decided to unify the unit in accordance with the applicable requirements of scientific research and in accordance with the suggestions both of reviewers.

Point 4: Table 1: it is very long and difficult to read. Can it be represented e.g. as a heatmap?

Response 4:

We changed Table 1. Now it is Figure 2 and the data was presented in the form of a heatmap. We hope the revised version of Table 1 will be satisfactory.

Point 5: Discussion is very long record of ample facts from interactions of cyanobacteria with each other and microalgae. it should be shortened e.g. by converting some fact into a figure e.g. with a matrix. The species (maybe with the corresponding references) can be listed on X and Y dimensions and the interaction type (++, +- or --) can be indicated on the intersections.

Response 5:

The authors agreed with the reviewer and extended mentioned part of the discussion.

Point 6: LL438-439: incomplete sentence?

Response 6: The mistake, has been corrected, authors apologize for this error.

Revised paragraph (Lines 438-439):

Cyanobacteria and algae belonging to the phytoplankton community were counted and identified using an inverted microscope (Nikon, Japan).

Reviewer 2 Report

General comments

The article’s topic is interesting and important in the context of danger for phytoplankton bioceonosis from side of toxic Cyanobacteria. Unfortunately, in the present form the manuscript is very weak and requires major changes. The presentation of the results and, most of all, the language of the paper, especially in the chapter 'Results' is very much in a style of a student’s presentation during a seminar (i.e. “in fig. X we can see this and this, and in table X this and that”), and not in a professional, scientific language which may be accepted in scientific journals. This is a very big drawback of the paper. The results are presented in an unclear way: sometimes the difference is given in percent, at other times as the multiplication of the difference. This is confusing. It is difficult to follow, let along interpret. The graphic way of presenting the results is also weak.

The aim of the study is imprecise, the hypothesis are lacking as well as specific Conclusion. The title is inadequate to the content of the work and must be changed. An article with a very similar titles has been published in 2017, it seems by the same author.

In the Introduction the author elaboration is much too wide in scope. I agree that the salt water environment has to be introduced, but such an introduction should not be excessive, it’s not the core part of the article. It is unclear why the author use the term “a natural plankton assemblage” with regard to lakes. The author do not explain it. The distinction into natural and non-natural (artificial) water bodies is clear, others – not really.

In many places in the Material and Methods, and Results some basic knowledge deficiency are manifest, which impact the whole study and the interpretation of the results. For example, the author writes that during the study period, the samples were taken at the depth of 1 m and 10 m, which correspond to the layers of epi- and metalimnion. about the seasonal changes in the thermal structure of the lakes, also not on the thickness of metalimnion. It is also unclear why in Table 2 the results seem to be referring to the ‘depth 0-1 m’ and not 1 m, that is to the depth where the samples were collected.

The article lacks a central idea, despite the apparent rich experience of the author in a phycological and hydrobiological research. The criteria for the selection of lakes for the study is unknown. The author offers a bundle of data on morphometry, percentage share of ‘woodland’ in the catchment and mentions the hydrological connection between the lakes (‘lake have one outflow which supplies water to the lake’). It is unfortunately not illustrated by an image, although the author included a large image of the study’s location (Fig. 3). The exact data on the trophic state of the lake water. Author must make an effort to open up their study to a comparative analysis, beyond its current narrow focus on a specific local area and its problems (currently, it is a typical idiosyncratic study).

Material and methods: This part is particularly incomplete and full of shortcomings. In case of determination of the allelopathic activity of Synechococcus sp. as source was given publication No. 22. The study can be cited, but in the section Material and Methods the author would also have to introduce the method. There is a nearly total lack of data about abiotic part of research, especially about the physicochemical research. Correctly, the author should name the used equipment for the field measurements as well laboratory methods of analysis for each parameter. If the authors use a method compatible with the standard methods for freshwaters, they must specify the international signature. Every scientist should know that the description of the methods is very important! In their absence, it is impossible for other researchers to compare those results with their own research in other parts of the globe. Another weak side of the article is wrong units of parameters (not according to SI system).

Discussion: The style of this section reminds rather the part of an overview paper than an actual discussion of own research results. In the present form, the discussion is not acceptable.

Reference list is a weak part of the manuscript too: there are many articles by the same author (Śliwińska-Wilczewska (9)); 2 books of elementary knowledge; 2 papers unrelated to the subject of the present article; journals names are given in full instead of abbreviations.  

The captions of all figures and tables are incomplete. The text of a caption must contain a complete set of information so that the reader can understand the content without the help of the paper.

The English needs improvement by a native English speaker in terms of both style and structure. The manuscript needs to be thoroughly checked by a native speaker in order to improve readability of the text and to facilitate interpretation of the research findings.

To sum up, the entire manuscript must be significantly improved. Results and Discussion needs reorganization. Clear aim and hypotheses must be provided, and the discussion needs a guiding argument. Generally, manuscript must be reorganized, revised and some sections must be written anew.

Particular comments (36) and corrections – see the manuscript.

Major revisions must be implemented before the manuscript can be considered for publication.

Author Response

Response to Reviewer 2 Comments

The authors would like to thank Reviewer 2 for her/his helpful and constructive comments that will greatly contribute to the improvement of our paper. We have studied your comments carefully and have made a correction. We really hope the revised version will be satisfactory. All the modifications in the manuscript are marked in blue color.

General comments

Point 1: The article’s topic is interesting and important in the context of danger for phytoplankton bioceonosis from side of toxic Cyanobacteria. Unfortunately, in the present form the manuscript is very weak and requires major changes. The presentation of the results and, most of all, the language of the paper, especially in the chapter 'Results' is very much in a style of a student’s presentation during a seminar (i.e. “in fig. X we can see this and this, and in table X this and that”), and not in a professional, scientific language which may be accepted in scientific journals. This is a very big drawback of the paper. The results are presented in an unclear way: sometimes the difference is given in percent, at other times as the multiplication of the difference. This is confusing. It is difficult to follow, let along interpret. The graphic way of presenting the results is also weak.

Response 1: The authors thank for all comments. They tried to include them all and edit the article that it took the appropriate form, had a clear message and scientific language. The authors added charts to better visualize research results.

Point 2: The aim of the study is imprecise, the hypothesis are lacking as well as specific Conclusion. The title is inadequate to the content of the work and must be changed. An article with a very similar titles has been published in 2017, it seems by the same author.

Response 2: The aim of the study, the hypothesis and specific Conclusion were added. The title was changed.

Point 3: In the Introduction the author elaboration is much too wide in scope. I agree that the salt water environment has to be introduced, but such an introduction should not be excessive, it’s not the core part of the article. It is unclear why the author use the term “a natural plankton assemblage” with regard to lakes. The author do not explain it. The distinction into natural and non-natural (artificial) water bodies is clear, others – not really.

Response 3: The scope of the Introduction has been reduced as suggested by the Reviewer. The use the term "a natural plankton assemblages" was inappropriate and has been removed throughout the article.

Point 4: In many places in the Material and Methods, and Results some basic knowledge deficiency are manifest, which impact the whole study and the interpretation of the results. For example, the author writes that during the study period, the samples were taken at the depth of 1 m and 10 m, which correspond to the layers of epi- and metalimnion. about the seasonal changes in the thermal structure of the lakes, also not on the thickness of metalimnion. It is also unclear why in Table 2 the results seem to be referring to the ‘depth 0-1 m’ and not 1 m, that is to the depth where the samples were collected.

Response 4: The concept of epilimnion and metalimnion was not used correctly, therefore it was removed from the entire article.

Point 5: The article lacks a central idea, despite the apparent rich experience of the author in a phycological and hydrobiological research. The criteria for the selection of lakes for the study is unknown. The author offers a bundle of data on morphometry, percentage share of ‘woodland’ in the catchment and mentions the hydrological connection between the lakes (‘lake have one outflow which supplies water to the lake’). It is unfortunately not illustrated by an image, although the author included a large image of the study’s location (Fig. 3). The exact data on the trophic state of the lake water. Author must make an effort to open up their study to a comparative analysis, beyond its current narrow focus on a specific local area and its problems (currently, it is a typical idiosyncratic study).

Response 5: The lakes from which materials to biological analysis (allelopathy) were collected are research objects of a large international project. They were chosen so to represent different trophic state and meet the criterion for the formation of appropriate sediments. Descriptions and map corresponding to lake description have been unified.

Point 6: Material and methods: This part is particularly incomplete and full of shortcomings. In case of determination of the allelopathic activity of Synechococcus sp. as source was given publication No. 22. The study can be cited, but in the section Material and Methods the author would also have to introduce the method. There is a nearly total lack of data about abiotic part of research, especially about the physicochemical research. Correctly, the author should name the used equipment for the field measurements as well laboratory methods of analysis for each parameter. If the authors use a method compatible with the standard methods for freshwaters, they must specify the international signature. Every scientist should know that the description of the methods is very important! In their absence, it is impossible for other researchers to compare those results with their own research in other parts of the globe. Another weak side of the article is wrong units of parameters (not according to SI system).

Response 6: The chapter has been corrected and supplemented with necessary data.

Point 7: Discussion: The style of this section reminds rather the part of an overview paper than an actual discussion of own research results. In the present form, the discussion is not acceptable.

Response 7: The authors re-edited Discussion to make it more relevant to the results of their own research.

Point 8: Reference list is a weak part of the manuscript too: there are many articles by the same author (Śliwińska-Wilczewska (9)); 2 books of elementary knowledge; 2 papers unrelated to the subject of the present article; journals names are given in full instead of abbreviations.  

Response 8: The authors have completed the literature. They also verified the legitimacy of citing two controversial articles and replaced them by other items. Articles by Śliwińska-Wilczewska contain one of the most recent results regarding allelopathy of picocyanobacterium Synechococcus sp. However, the result of studies by other scientists are also cited.

Point 9: The captions of all figures and tables are incomplete. The text of a caption must contain a complete set of information so that the reader can understand the content without the help of the paper. The English needs improvement by a native English speaker in terms of both style and structure. The manuscript needs to be thoroughly checked by a native speaker in order to improve readability of the text and to facilitate interpretation of the research findings.

Response 9: The captions of all figures and tables have been verified and corrected. The article has been checked by a native speakers.

Point 10: To sum up, the entire manuscript must be significantly improved. Results and Discussion needs reorganization. Clear aim and hypotheses must be provided, and the discussion needs a guiding argument. Generally, manuscript must be reorganized, revised and some sections must be written anew.

Response 10: The manuscript has been completely rewritten, especially Results and Discussion as suggested by the Reviewer.

Particular comments (36) and corrections – see the manuscript.

Major revisions must be implemented before the manuscript can be considered for publication.

Particular comments:

1) Please change/correct of the title. Item 22 is "First record of allelopathic activity of the picocyanobacterium Synechococcus sp. on a natural plankton community" 2017. 'First evidence' and 'first record' are in practice synonyms.

Response 1:

Thank you for that comment. We think these two titles: “The First Evidence of Allelopathy in Freshwater Picocyanobacterium Synechococcus sp.” and “First record of allelopathic activity of the picocyanobacterium Synechococcus sp. on a natural plankton community” are different, however, we decided to change the title to “The importance of allelopathic picocyanobacterium Synechococcus sp. on the abundance, biomass formation, and structure of phytoplankton assemblages in three freshwater lakes”.

2) What you mean in term "natural plankton assemblages"? Generally in natural waterbodies all phytoplankton bioceonosis are natural.

Response 2:

We corrected this aspect.

3) General remark to the manuscript:

In scientific works, the authority for a binomial name is usually given, at least when it is first mentioned.

Response 2:

We agree with the reviewer's comment. We corrected this aspect in whole manuscript.

4) freshwater or inland freshwater lakes

Response 4:

We changed Keywords according to reviewers suggestions.

5) I suggest only 'Phytoplankton', because zooplankton was not taken into consideration.

Response 5:

We changed Keywords according to reviewers suggestions.

6) Eliminate citation of book 1-2. Add papers!

Response 6:

We eliminated citation of book 1-2.            

7) General remark: Please concern on freshwater! Marine environment may be taken into consideration but not in so large range as now.

Response 7:

We corrected this aspect.

8) Line 63-74: weak signification for the topic of manuscript. Please reduct to 1 sentence.

Response 8:

According to reviewer suggestion, this paragraph was reduced.

9) Repetition (see line 418-422)

Response 9:

We deleted this repetition.

10) maybe only 'phytoplankton'

Response 10:

We corrected this aspect.

11) In the place for each of the lake add abreviation (ŁL, RL, ZL) and used them in the text, figures, tables and captions.

Response 11:

We agree with the reviewer's comment. We corrected this aspect in whole manuscript.

12) Please note, that the 'most numerous' may be only one.

Response 12:

We are sorry for this mistake. We changed this sentence.

13) Delete all mark (-) from tables S1-S4 (see Word file) and correct all caption of tables.

Response 13:

We deleted mark (-) and corrected caption of tables S1-S4.

14) Very complicated description!

Line 97-121: Instead of a huge number of percentage share values of phytoplankton groups in total biomass at different depth add table or figure, which ilustrated them in epi- and metalimnion.

List of dominant taxa in text is unnecessary because the reader can all see in Table S1.

Response 14:

We changed this paragraph. We also deleted the list of dominant taxa from MS. We did not add a table or figure with values, because in our opinion it would be a repetition of the results contained in Fig. 1. Therefore, we left the percentage data and entered the value of the biomass in brackets. We hope that this version will be satisfactory.

15) According to SI units must be changed all units in manuscript, also in tables and figures (see Instruction for Authors)

Response 15:

We corrected this aspect.

16) Please change orientation of the table, from vertical to horizontal.

Response 16:

Thank you for this comment however, we changed table 1 to fig. 2, according to the request of Reviewer 1, therefore, in our opinion, no change of table orientation will be needed here.

17) SI Units (International System of Units) should be used.

Response 17:

We do not fully understand this comment with respect to table 1. The number of algal cells per ml is a unit that is used in Toxins, so we would like to stay with this version of MS. We have changed the record a bit and Table 1 is now Fig. 2, we hope that this version will be satisfactory.

18) Change in whole manuscript

Response 18:

We corrected this aspect.

19) Line: 138-188

35 times the word ANOVA! In subchapter 4.4 is information that only ANOVA was analysed. Conclusion: elimination of 34 words ANOVA.

Response 19:

We are sorry for that. We eliminated the repetitive word “ANOVA” from the text.

20) Very poor illustration of results in figs. 1 and 2. First: Change range of Y axis (Number of cells) to max 150. Move legend to fig. A of the right column. Second: The color palette has low contrast. Change the colour palette: blue for Cyanoph., black for Bacill. and so on.

Response 20:

We corrected Fig. 1 and Fig. 2. We change range of Y axis (Number of cells) to 140 as well as moved legend to fig. A of the right column. We also tried to change the color of the Bacillariophyceae to black, but in our opinion it did not look good. That's why we've changed the contrast of the color palette. We hope that this version of Fig 1 and 2 will be satisfactory. Please note that at the request of Reviewer 2 we changed Tab1 to Fig.2, therefore the figures mentioned in this commentary are Fig 1 and Fig. 3.

21) Line: 378-388: Change to summary or specific conclusions.

Response 21:

We corrected this aspect.

22) Add Table with morphometrically parameters of study lakes as well as other data e.g. surface of catchment area, etc.

Response 22:

We added Table 2 with morphometrically parameters of study lakes as well as other data

22) Please shorten, e.g. Żabińskie Lake (area 41.6 ha, max depth 44.4 m) is ...

Response 22:

We corrected this aspect.

23) The description (394-413) should be changed (see notes to fig. 3)

Response 23:

We corrected this aspect.

24) Please specify of the type of catchment area: total or immediate? See: Klimaszyk et al. 2015 DOI: 10.1007/s12665-014-3682-y

Response 24:

We corrected this aspect.

25) Add more data about trophic state of lakes (according to TSI, Carlson 1977)

Response 25:

We added more data about trophic state of lakes.

26) Masurian Lakeland? From Stettin on the west?

Response 26:

We are sorry for this mistake. We corrected Fig. 3 according Reviewer’s suggestions.

27) Fig. 3 must be totally changed. It should be showed: local topographic map with location of all three lakes, area of Masurian Lakeland and then the map of Europe.  

Response 27:

We are sorry for this mistake. We corrected Fig. 3 according Reviewer’s suggestions.

28) Totally lack information about methods used to physicochemical analysis and measurement of abiotic features of water!!!

Response 28:

We are sorry for that. We added information about methods used to physicochemical analysis and measurement of abiotic features of water.

29) Lack data about the vertical thermal structure of lakes. How do I know, that at the depth 10 m was metalimnion? Samles were taken from upper, middle or down part of this layer. Please add figure with thermal gradient in lakes in all study months.

Response 29:

Water samples for analysis were taken from constant depths - 1 and 10 m. The manuscript contained a mental shortcut, the use of the term "metalimnion" was not entirely correct.

The attached figure shows that at a depth of 10 m in lakes Rzęśniki and Łazduny  there was the lower boundary of the metalimnion, however in the lake Żabińskie this may be a contentious issue.

That is way the term of metalimnion was removed from the manuscript.

Figure. Depth profiles of water temperature in A – Lake Żabińskie, B – Lake Łazduny, C – Lake Rzęśniki

30) Correct units of volume in whole manuscipt.

Response 30:

We corrected units of volume in whole manuscipt.

31) In case T and pH leave 1 decimal place, EC without decimal values.

Response 31:

We corrected this aspect.

32) If sampling was one time in month: how many results from the month were taken into calculation of mean value? Please clearly describe in Material and Methods.

Response 32:

Sampling was taken one time in month and one results from the month were taken to our analysis.

The form "mean" was used incorrect. We are sorry for this mistake. We corrected this aspect.

33) Please add a photo/-s from the experimental phase.

Response 33:

We added a new Figure 4 to MS.

34) From the same lakes as descibed above or different?

And at the same time?

Response 34:

A phytoplankton communities used for the experiment was taken from the same lakes and at the same time as phytoplankton for the biological analysis: species composition and biomass calculation.

35) How important is this information?

During sampling: in epilimnion or metalimnion?

Response 35:

Information about the temperature in this place of the manuscript is unnecessary. That is why it was removed from the text.

35) Change the names to abbreviations.

Response 35:

We corrected this aspect.

36) Items 69-70: Lack of substantive connection with the topic of manuscript.

Response 36:

We changed these citations.

Round 2

Reviewer 1 Report

The authors did an excellent job dealing with the comments and issues mentioned in my review. I believe that the paper can be published now.

Author Response

Response to Reviewer 1 Comments

The authors did an excellent job dealing with the comments and issues mentioned in my review. I believe that the paper can be published now.

Response:

We would like to thank you for your valuable comments and suggestions to improve the quality of our paper. Thank you very much for your nice cooperation.

Reviewer 2 Report

Answers and supplements are almost sufficient. Quality of manuscript now is significantly higher than earlier. Unfortunately, I haven't noticed some of the changes.
Currently, the work contains only minor shortcomings that require changes (see pdf file). Corrections will improve the readability and clarity of the scientific message.

Author Response

Response to Reviewer 2 Comments

Answers and supplements are almost sufficient. Quality of manuscript now is significantly higher than earlier. Unfortunately, I haven't noticed some of the changes.

Currently, the work contains only minor shortcomings that require changes (see pdf file). Corrections will improve the readability and clarity of the scientific message.

Response: The authors would like to thank Reviewer 2 for her/his valuable comments and suggestions to improve the quality of our manuscript. We apologize for not including all the corrections in the previous version of the MS. This was not a deliberate intention but only an oversight on our side. In this version, we have studied your comments carefully and have made a correction. We really hope the revised version will be satisfactory. All the modifications in the manuscript are marked in blue color.

  1. According to AlgaeBase binominal name of this species is: Cyanobium gracile Rippka & Cohen-Bazire; Scenedesmus quadricauda (Turpin) Brébisson. In case of all species in manuscript (except abstract) please complete the names as above.

Response 1:

We complete the species names according to the Reviewer's suggestion.

  1. There is no description of results in Fig. 4 showed

Response 2:

According to Reviewer comment we added this matter to the result section.

Added sentence (Lines 195-204):

“We also showed that different phytoplankton species responded differently to Synechococcus sp. exudates (Figure 4). It was found that Cyanophyceae from the genus Aphanothece, Limnothrix, Microcystis, Planktolyngbya, Pseudanabaena, Synechococcus, and Woronchininia, as well as Chlorophyceae and Charophyceae (Ankistrodesmus, Cosmarium, Dictyosphaerium, Pediastrum, Planktonema, and Scenedesmus) showed tolerance for allelopathic compounds produced and released by freshwater Synechococcus sp. in each lake. In our work we have shown that Synechococcus sp. also stimulated the growth of some species of Bacillariophyceae, especially from the genus Odontella and Stauroneis. On the other hand, Chroococcus and Lemmermaniella (Cyanophyceae), Sphaerocystis and Koliella (Chlorophyceae), as well as Achnanthes, Amphora, Gomphonema, and Nitzschia (Bacillariophyceae) were strongly inhibited by this picocyanobacterium.”

  1. Change: Biomass of other groups was clearly lower (Figure 1A).

Response 3:

We corrected this sentence.

  1. please change - see text

Response 4:

We changed it according to the Reviewer's suggestion.

  1. Line: 112-193
  2. Please eliminate ...-fold, change to a percentage share
  3. Please round percentage results below 100% to 5% (e.g. 6 = 5, 3 = 5, 72 = 50 68 = 70), and over 100% to 10% (e.g. 144 = 140, 175 = 170, 577 = 580).

Response 5:

We changed it according to the Reviewer's suggestion.

  1. All about results presented in Figure 4 should be moved to the Results section. Discussion is not a good place for results!

Response 6:

We changed discussion and moved some sentences to the Results section.

  1. Please change - see caption

Response 7:

We changed it according to the Reviewer's suggestion.

  1. Line: 282-295

Change paragraph to a form of conclusion of 3 points.

Response 8:

We changed this paragraph to a form of conclusion of 3 points.

Revised sentence (Lines 296-301):

“Our observations indicated that (i) some phytoplankton species may show tolerance for allelopathic compounds produced and released by freshwater Synechococcus sp., which may be the result of coevolution during their coexistence in some freshwater ecosystem, (ii), mutual stimulation of picocyanobacteria may explain their high abundances in the summer season in some freshwater reservoirs, (iii) the allelopathic effect may be dependent on the specificity of the target group and season.”

  1. Please eliminate rectangular box on the map of Poland

Response 9:

We eliminated rectangular box on the map of Poland.

  1. change: the base of chlorophyll-a

Response 10:

We corrected this aspect.

  1. Change: status of

Response 11:

We changed it according to the Reviewer's suggestion.

  1. Add: state of both

Response 12:

We corrected this aspect.

  1. Please correct

Response 13:

We changed it according to the Reviewer's suggestion.

  1. Please correct (2x)

Response 14:

Thank you for this comment. We have not seen this error before. We corrected this aspect.

  1. please add: membrane

Response 15:

We added the word “membrane” to MS.